# Intratumoral Influenza Vaccine Administration Attenuates Breast Cancer Growth and Restructures the Tumor Microenvironment through Sialic Acid Binding of Vaccine Hemagglutinin

**DOI:** 10.3390/ijms25010225

**Published:** 2023-12-22

**Authors:** Preston Daniels, Stefanie Cassoday, Kajal Gupta, Eileena Giurini, Malia E. Leifheit, Andrew Zloza, Amanda L. Marzo

**Affiliations:** 1Department of Internal Medicine, Division of Hematology and Oncology, Rush University Medical Center, Chicago, IL 60612, USA; preston_daniels@rush.edu (P.D.); malia_e_leifheit@rush.edu (M.E.L.); andrewzloza@gmail.com (A.Z.); 2Department of Microbial Pathogens and Immunity, Rush University Medical Center, Chicago, IL 60612, USA; stefanie_l_cassoday@rush.edu; 3Department of Surgery, Rush University Medical Center, Chicago, IL 60612, USA; kajal_gupta@rush.edu (K.G.); eileena_f_giurini@rush.edu (E.G.)

**Keywords:** breast cancer, influenza vaccine, immunotherapy, tumor microenvironment, sialic acid, hemagglutinin

## Abstract

Breast cancer continues to have a high disease burden worldwide and presents an urgent need for novel therapeutic strategies to improve outcomes. The influenza vaccine offers a unique approach to enhance the anti-tumor immune response in patients with breast cancer. Our study explores the intratumoral use of the influenza vaccine in a triple-negative 4T1 mouse model of breast cancer. We show that the influenza vaccine attenuated tumor growth using a three-dose intratumoral regimen. More importantly, prior vaccination did not alter this improved anti-tumor response. Furthermore, we characterized the effect that the influenza vaccine has on the tumor microenvironment and the underlying mechanisms of action. We established that the vaccine facilitated favorable shifts in restructuring the tumor microenvironment. Additionally, we show that the vaccine’s ability to bind sialic acid residues, which have been implicated in having oncogenic functions, emerged as a key mechanism of action. Influenza hemagglutinin demonstrated binding ability to breast cancer cells through sialic acid expression. When administered intratumorally, the influenza vaccine offers a promising therapeutic strategy for breast cancer patients by reshaping the tumor microenvironment and modestly suppressing tumor growth. Its interaction with sialic acids has implications for effective therapeutic application and future research.

## 1. Introduction

Breast cancer continues to pose a significant clinical challenge to the health of women worldwide due to relapse and resistance to conventional therapeutic approaches. Devising new clinical strategies is a necessity for improving the outcomes of women living with treatment-resistant breast cancer [1,2,3]. The immunosuppressive tumor microenvironment (TME) of breast cancer has been identified as a central bottleneck to addressing tumor progression and therapeutic resistance against modern targeted immunotherapies. Without adequately addressing the various molecular and cellular components that support tumor growth and suppress anti-tumor immune responses, there is limited potential for targeted cancer immunotherapies to have meaningful success for this challenging breast tumor type [4,5,6]. Therefore, altering the TME with effective interventions is crucial in tipping the balance from a tumor-promoting microenvironment to a tumor-killing one and ultimately transforming cancer growth and treatment outcomes.

Pathogens have historically been recognized as cancer-causing agents [7,8]. However, their pleiotropic nature and potential use in oncology have become more appreciated over time [9]. Beyond their well-researched direct oncolytic capabilities, viruses and their derivative components offer a unique toolkit to modulate the host immune response, thereby reshaping the TME to induce tumor immunity. The convergence in the evolutionary immune response to kill pathogens directly overlaps with potential cytotoxic mediators that can have positive anti-tumor immune effects in cancer cell killing and reshaping the TME [10]. This convergence and potential of co-opting anti-viral immunology for cancer treatment has been demonstrated with the use of Toll-Like Receptor (TLR) agonists in a clinical oncology setting, producing early signs of benefit for use of imiquimod against breast cancer skin metastasis [11,12]. In addition, preclinical work has demonstrated the capacity of various clinically approved vaccines to alter and improve anti-tumor immune responses in several cancers [13,14,15,16]. This is best supported by the approved use of the vaccine Bacillus Calmette-Guerin (BCG) in the treatment of non-muscle invasive bladder cancer (NMIBC) [17,18].

While less characterized than BCG in a cancer setting, the influenza virus has distinctive biological properties that further support investigating its uses against breast cancer. Previous studies have shown that the influenza virus is tropic to tumor cells and that minor genetic edits can transform the virus to an effective oncolytic drug [19]. Furthermore, previous work by our group has demonstrated the efficacy of the FDA-approved seasonal influenza vaccine in a melanoma mouse model by facilitating CD8+ T cell infiltration into the tumor [13]. This finding has also been demonstrated clinically in colon cancer, where the influenza vaccine injected into tumor tissues facilitated T cell infiltration into the tumor microenvironment upon analysis after post-treatment resections [20].

The clinically approved influenza vaccine is formulated to comprise predominately wild-type influenza hemagglutinin (HA) proteins of three or four influenza strains as the vaccine’s main antigenic target. This formulation has been a cornerstone of combating seasonal flu outbreaks for decades [21]. In the active form of the virus, the essential surface protein, HA, facilitates viral entry through binding sialic acids expressed on tissues with high sialic acid expression [22]. This interaction, while central to the virus’s pathogenicity, offers opportunities in the realm of therapeutic oncology.

Sialic acids play multifaceted roles in healthy cellular interactions and signaling. However, hyperexpression of sialic acids in breast cancer has been associated with pro-oncogenic functions, including promoting tumor cell invasion, metastasis, and angiogenesis [23,24,25,26]. Furthermore, sialic acids can exert immune inhibitory effects through immunoreceptor tyrosine-based inhibitory motif (ITIM) signaling and sialic acid-binding immunoglobulin-like lectins (Siglecs). This signaling blunts anti-tumor immune responses and can create an immunosuppressive TME [27,28,29]. In light of these oncogenic and immune-suppressive roles, strategies aimed at modulating the sialic acid axis hold significant therapeutic promise and have recently emerged as an area of interest for further investigation. By administering influenza HA intratumorally, the potential exists to bind and block intratumoral sialic acids, thereby reshaping the TME to favor enhanced immune activity and robust anti-tumor responses. The multivalent composition of the influenza vaccine, encompassing HA from multiple influenza strains, offers a comprehensive approach to target the diverse conformations of sialic acids that may exert suppressive or oncogenic functions within the tumor [30,31,32]. With the recent emergence focusing on the sialic acid pathway in cancer, the potential to repurpose a well-established clinical agent such as the influenza vaccine that has an exceptional safety profile at a low per unit cost presents a promising therapeutic strategy and direction of research for modulating this pathway in breast cancer.

## 2. Results

### 2.1. Intratumoral Administration of Flu Vaccine Reduces 4T1 Breast Tumor Growth

Utilizing the well-validated 4T1 syngeneic triple-negative breast cancer model, we evaluated the therapeutic potential of intratumorally administering flu vaccines for use against breast cancer. Using a repeat dosing schedule (Figure 1A), we determined that the influenza vaccine (FluVax) significantly, although modestly, curtailed tumor growth compared to PBS control (Figure 1B). When examining the treatment potential of reduced dosing, we confirmed that three consecutive daily doses of FluVax had maximal anti-tumor efficacy. However, the therapeutic effectiveness of FluVax was decreased when the dosing frequency was reduced (Figure 1D). This phenomenon of reduced dosing having reduced anti-tumor efficacy was independent of temporal considerations of every-other-day dosing for two doses. In support of our data, it has been reported that less frequent dosing of BCG in bladder cancer results in poorer outcomes and higher rates of recurrence both pre-clinically and clinically [16,33].

### 2.2. Pre-Existing Vaccination Does Not Hinder Efficacy of Subsequent Intratumoral Influenza Vaccine Administrations

To understand the translational potential of our findings, we investigated the impact of prior vaccination; thus, we proportionately increased the age in the 4T1 mouse model on tumor progression and the therapeutic efficacy of repeat intratumoral influenza vaccine administrations. Data from tumor growth trajectories demonstrated that prior vaccination with the influenza vaccine prior to tumor induction did not alter the inherent growth dynamics of the tumors compared to vaccine-naive mice (Figure 2A,B). This implies lack of overlap in antigenic responses to the vaccine and cancer, as well as that the influenza vaccine functions as an adjuvant or immune modulator, not an antigenic source. This aligns with what we know about influenza hemagglutinin, as it is a poor CD8+ T cell antigen in humans [34]. When examining the impact of this more translationally relevant model in a treatment paradigm, mice treated with three consecutive daily intratumoral doses post-vaccination (Vaxed-FluVax ×3 continue to demonstrate a marked reduction in tumor volume (Figure 2C,D). This demonstration is essential to indicate that existing anti-HA immune responses, which would be expected in most patients in a clinical setting, do not affect FluVax anti-tumor benefits within the TME [35]. To further maximize the therapeutic effect in this model, we increased the treatment frequency to seven daily doses (FluVax ×7). However, we saw only modest non-significant improvements in response following this treatment regimen, demonstrating diminishing returns of additional doses after three daily doses (Figure 2D). This increased dosing regimen did not, however, affect mouse body weight, further highlighting the safety of this agent and approach (Figure 2F). Hemagglutination assays validated the efficacy of prior vaccination, with clear agglutination in the vaccinated mice blood samples, confirming seroconversion following our vaccination procedure prior to tumor induction (Figure 2E). This assay additionally highlighted the retained ability of hemagglutinin within the influenza vaccine to bind sialic acid.

### 2.3. Intratumoral Influenza Vaccine Administration Reshapes the 4T1 Tumor Microenvironment

To characterize the impact of repeat administration of the influenza vaccine on the 4T1 tumor microenvironment, RNA expression was evaluated using the Murine nCounter^®^ Immune Exhaustion Panel. Mice were subjected to three doses of intratumoral influenza vaccine treatment and monitored for growth before being sacrificed for tumor tissue, from which RNA was extracted (Figure 3A,B). Analysis of the transcriptional landscape induced by the vaccine demonstrated a discernable shift in differentially expressed genes in the 4T1 tumor microenvironment post-vaccination. The volcano plot (Figure 3C) and heatmap (Figure 3D) illustrate these transcriptional alterations. There was a substantial increase in IL-24, an IL-4-induced cytokine with demonstrated targeted anti-breast cancer activity [36,37]. Furthermore, there was a noticeable decrease in lipoprotein lipase (LPL), a metabolic enzyme implicated in breast cancer proliferation and metastasis [38,39]. Finally, pivotal effector cell immune genes such as ICOS, PD-L2, and TIGIT showed strong upregulation. This is further supported by concurrent increases in leukocyte population gene sets (CD45) being upregulated in cell profiling analysis derived from this gene expression panel, as well as by inferred increases in tumor-infiltrating lymphocytes with a skewed Th1 T cell and dendritic cell (DC) profile compared to PBS weeks after therapeutic administration (Figure 3E). While there were outliers of differential response and immune composition in each group (PBS and FluVax), leukocyte upregulation in the PBS group looks to be primarily driven by increases in Tregs, without concurrent increases in Th1 cells as seen in FluVax treatment.

### 2.4. Intratumoral Influenza Vaccine Upregulates IL-4, Which in Part Mediates Anti-Tumor Effects

Repeating our findings, as schematically represented in Figure 4A, we took tumor samples from the mice and homogenized tumor tissue to conduct a Luminex multiplex in order to elucidate any cytokine shifts facilitated by the intratumoral administration of the influenza vaccine (Figure 4B). Upon analysis of the immune mediators of the tumor, we saw significant downregulation of pathological inflammatory signaling of IL-6 and IL-1beta, which are each implicated in breast cancer tumor growth and progression (Figure 4C,D) [40,41]. IL-4 expression, however, was the only differentially expressed cytokine between a single and three repeat administrations of influenza vaccine, implying its potential key role in mediating the anti-tumor response (Figure 4C–E). Corroborating this data was the upregulation of IL-24 gene expression in Figure 3C, a known downstream signal of IL-4 signaling [42,43,44]. To examine the anti-tumor impact of increased IL-4 production, we blocked IL-4 intratumorally using an IL-4 blocking antibody (Figure 4F). Upon blocking IL-4 intratumorally, all anti-tumor benefit of the influenza vaccine was eliminated (Figure 4G,H). This demonstrated the importance of this cytokine within this treatment paradigm and model. These findings collectively indicate that FluVax may elicit its anti-tumor effects in part through the upregulation of IL-4 and its downstream effects, potentially including IL-24 upregulation.

### 2.5. Sialic Acid Binding Ability Mediates the Anti-Tumor Activity of the Influenza Vaccine

Having demonstrated the retained ability of the influenza vaccine to bind sialic acid in the hemagglutination assay of vaccinated mice in Figure 2E we wanted to evaluate this potential interaction intratumorally. First, we heat-inactivated the influenza vaccine and demonstrated a loss of sialic acid binding ability following high heat exposure, as demonstrated by loss of agglutination (Figure 5B). Following this, mice were given repeat administrations of unmodified influenza vaccine or heat-inactivated vaccine for three doses (Figure 5A). Mice receiving the unmodified vaccine showed the expected benefit in anti-tumor response, while the group receiving heat-inactivated vaccine did not show this benefit (Figure 5C). To understand whether this loss of effect was specifically due to loss of binding activity, we sought to block vaccine hemagglutinin binding without modifying the vaccine hemagglutinin structure. This allowed us to rule out mechanisms other than sialic acid binding that could also be related to the heat inactivation procedure and loss of wild-type protein conformation. We blocked the sialic acid binding of the influenza vaccine by titrating exogenous synthetic sialic acid Neu5Ac into the vaccine composition and monitored the effects by hemagglutination assay. A clear inhibition of hemagglutination activity was observed with a mixture of exogenous Neu5Ac residues, indicative of blocked hemagglutinin binding sites (Figure 5E).

Interestingly, the exogenous Neu5AC showed cytotoxic effects on red blood cells when administered at high concentrations exceeding the titrated balance of blocking sialic acid binding of hemagglutinin (Figure 5E). Using the titrated mixture of Neu5AC and FluVax, mice underwent treatment to assess the impact of blocking the flu vaccine’s sialic acid binding activity within the tumor (Figure 5D). Upon monitoring tumor growth using this mixture, it was observed that while the group treated with the flu vaccine or Neu5AC independently had anti-tumor effects intratumorally, when titrated to a composition that eliminated sialic binding, they each lost anti-tumor efficacy. This underscores sialic acid binding as a key mechanism of action in the influenza vaccine’s anti-tumor application, and indicates sialic acid expression as a promising biomarker for clinical application.

### 2.6. Tumoral Sialic Acid Expression Modulates Influenza Hemagglutinin Binding

Our investigations into sialic acid binding highlighted the potential necessity of retained hemagglutinin binding ability. To further understand this dynamic in the 4T1 breast cancer model, we first measured the expression pattern of key sialic acid linkages that are known to be tropic for human influenza strains [31,32]. Using the biotinylated lectins SNA and MAL II, we observed by flow cytometry a high expression of α-2,3 and α-2,6 sialic linkages with a modest elevation in the expression of the latter (Figure 6A). To understand the translational relevance of this finding, we examined the profile of a human breast cancer cell line, MDA-MB-231. We saw a similar elevation of expression, though with a more balanced profile of these sialic acid linkages (Figure 7A). To understand the ability of hemagglutinin to bind these tumor cells, we utilized His-tagged hemagglutins of the same strains of influenza that comprise the seasonal influenza vaccine and used a sialyltransferase inhibitor P-3FAX-Neu5Ac to inhibit sialic acid expression in these cells [45]. In both 4T1 and MDA-MB-231, we showed that P-3FAX-Neu5Ac was capable of downregulating both α-2,6 (Figure 6C and Figure 7C) and α-2,3 (Figure 6D and Figure 7D) linkages, with more significant downregulation of the latter in MB-MDA-231 (Figure 7D). Using equal distribution of the various strains of the his-tagged hemagglutinin, as is utilized in the clinical formulation of the influenza vaccine, we demonstrated the binding ability of influenza hemagglutinin to bind to tumor cells in both murine (Figure 6D) and human (Figure 7D) breast cancers. This ability of our tagged hemagglutinin mixture to bind to tumor cells was then downregulated when used with cells cultured with the sialyltransferase inhibitor P-3FAX-Neu5Ac (Figure 6E and Figure 7E). This demonstrates that tumoral hyperexpression of sialic acids plays a key role in modulating the binding dynamics of influenza hemagglutinin, which we have shown to be critical to its efficacy as an anti-tumor therapy (Figure 5).

## 3. Discussion

Historically, viruses have been viewed through the lens of pathogenicity both as infectious and oncogenic agents. However, recent advances in oncolytic virotherapy have shifted this perspective, highlighting the potential of viruses as therapeutic agents in cancer treatment [9]. Furthermore, various pathogen vaccines and isolated pathogen components have been reimagined as immunotherapies in the fight against cancer [13,14,15,16]. Our study centered on the therapeutic potential of intratumoral administration of the influenza vaccine against breast cancer. We have demonstrated a set of findings that underscore the vaccine’s underlying anti-tumor mechanisms that could inform future research and clinical application of this approach.

The demonstrated anti-tumor effects of the influenza vaccine were statistically significant; however, they were modest in terms of total tumor reduction in this model. Nonetheless, the anti-tumor efficacy of this approach is in line with that of other promising single agent immunomodulatory approaches in the resistant and highly aggressive 4T1 breast cancer model [16,46,47,48]. Notably, this parallel response can be achieved with a clinically approved agent with a highly favorable safety profile. Considering that monotherapy is inadequate for curative outcomes in this model of checkpoint refractory TNBC, the favorable safety profile of this therapeutic strategy becomes essential when considering its integration into combination regimens. This is particularly critical because combinatorial approaches may give rise to synergistic toxicities, potentially impeding the effective application of such strategies. Furthering this combination utility, we have generated data characterizing this monotherapy approach with mechanistic data that can be utilized to inform rational combination strategies of future investigations.

The influenza vaccine’s inherent ability to bind to sialic acid residues is central to our observations. Sialic acids are crucial in various physiological and pathological processes, including cell–cell interactions, immunity, and cancer progression. Aberrant sialylation has been associated with tumor invasiveness, metastasis, and immune evasion [22,23,24,25,26]. While the implications of this pathway are promising as a therapeutic target there exist significant challenges in practice. This is a diverse and redundant pathway that expresses a multitude of sialic acid conformations and is regulated by various intracellular mediators. Extracellularly, there is a wide set of immune-inhibiting Siglecs that, upon binding the sialic acid for which they have a specific affinity, are inhibited in their function through their downstream intracellular ITIM signaling [49,50,51]. Considering the amount of diversity and lack of fidelity, this poses a challenge to conventional targeted therapies. However, using the diverse set of strains of influenza hemagglutinin comprised within the influenza vaccine, which inherently have differential affinities, it is possible to target multiple sialic acid conformations and modulate this pathway in an effective and cost-efficient manner. This provides substantial promise in light of our finding on the vaccine’s ability to bind sialic acid residues in this paper. This finding holds meaningful therapeutic implications, potentially allowing for the use of well-defined biomarkers such as sialic acid expression for patient selection in order to maximize immune activity and anti-tumor responses.

An often-underappreciated element of the therapeutic application in early therapeutic development is defining the optimal dosing strategies that should be implemented. Defining the intricate dynamics between dose, timing, and therapeutic efficacy is a cornerstone of good translational research. This has even been implicated as a critical component in the application of BCG for the treatment of bladder cancer, where the dosing frequency was deemed vital in reducing the rate of relapse [33]. Our findings from Figure 1 and Figure 2 emphasize the importance of a dosing regimen in the application of the influenza vaccine, where the optimal application consisted of three consecutive daily doses of FluVax. This regimen was our most potent strategy, without being overly demanding, as additional doses conferred little to no benefit (Figure 2D). The differential responses observed between one- or two-dose regimens and three doses highlight the nuanced importance of proper therapeutic application to maximize anti-tumor effects.

When we think about the translational relevance of this application, we are confronted with the reality that, inevitably, all breast cancer patients likely have an existing baseline of anti-influenza immune response [35]. This could be from prior infection or from vaccination; however, as influenza is a common illness, it is critical to characterize this response in a representative model. Using a vaccinated model allowed us to examine the effect of prior vaccination. It additionally provided the benefit of enabling the mice to age into maturity, providing a better representation of the clinical breast cancer population. Confirming efficacy in this model with our thrice-intratumoral injection approach further validated our strategy for future clinical relevance.

The observable restructuring of the 4T1 tumor microenvironment post-intratumoral influenza vaccine administration provides further insight into the mechanism at play in this approach. In our study, the marked upregulation of IL-4, traditionally associated with T-helper 2 (Th2) immune response and not an anti-tumor response, was an unexpected finding. However, while classical characterization of IL-4 is not that of a favorable cytokine in oncology, there is evidence of contextual benefit to IL-4 in certain oncology settings [52,53,54,55]. This is supported by our immune gene profiling work, which highlights an upregulation of dendritic cell and Th1-type immune gene sets at time points approximately three weeks post-treatment (Figure 3E). This is in comparison to our IL-4 observations, which characterized the early changes in molecular mediators (Figure 4), which would infer that the influenza vaccine triggers IL-4 secretion that facilitates downstream enhanced dendritic cell and Th1-type immune activity, ultimately resulting in attenuated tumor growth. Although potentially counterintuitive to classical immuno-oncology paradigms, this function of IL-4 enhancing Th1 responses through dendritic cell activity is in line with the observed mechanisms of IL-4 anti-tumor activity in other published literature [52].

Furthermore, in light of the relationship of IL-4 as an upstream activator of IL-24 expression, which we saw substantially upregulated and is a cytokine with demonstrated anti-breast cancer activity, we believe that the influenza vaccine is facilitating a highly unique shift in the immune landscape within the tumor [36,37,42,43,44]. This dynamic shift to the resulting tumor microenvironment is essential to consider when evaluating potential combination strategies, particularly with immunotherapies. Such unique alterations in the cytokine milieu post-FluVax treatment will require further empirical exploration of logical synergistic combinations with the influenza vaccine in order to maximize therapeutic efficacy. Our findings revealed a notable dichotomy in the immune response within the tumor microenvironment post-vaccine administration beyond the unique observation of IL-4 activity. Alongside the upregulation of immunostimulatory markers such as IL-24, IL-12, TNF, ICOS, IL2ra, and NFAT, there is a concurrent increase in immunosuppressive genes, including IDO1 and 2, VEGFA, and both PD-L2 and TIGIT. Targeted therapies specifically aimed at these immunosuppressive pathways upregulated with the use of the influenza vaccine could potentially enhance the vaccine’s anti-tumor efficacy by counteracting any vaccine-induced immunosuppressive signals within the tumor microenvironment while preserving the immune activation. However, considering the unique characteristics of IL-24 to induce apoptosis, inhibit angiogenesis, increase T cell fitness, and suppress metastatic potential in breast cancer and other tumor types, there are likely ample avenues for potential synergistic therapies [36,37,56,57,58].

Adding our observations in this study to prior preclinical work in melanoma and clinical observations in colorectal cancer demonstrates the versatility of the influenza vaccine and positions it as a unique therapeutic agent in oncology [13,20]. The unique profile of being immunostimulatory while maintaining an exceptional safety profile provides a low-risk opportunity for potential combinations that are capable of being investigated and integrated into a wide range of cancer treatment paradigms. Furthermore, the broad availability and large-scale annual production of influenza vaccines provide unique access to this investigational application while limiting financial burdens, which can often be of chief concern when using combination approaches. 

While our findings are promising, a deeper exploration of the molecular mechanisms underlying intratumoral influenza vaccine treatment is warranted. While different seasonal strains and compositions of the influenza vaccine have shown efficiency in other models, further research could be carried out to examine the most critical individual strains within the vaccine. In this same vein, additional strains of influenza could be investigated that are not typically vaccinated against. While less translationally relevant, this could be an exciting avenue for further investigation, as cancers have been known to express conformations of sialic acid not found in healthy human tissue, only in other mammals [30]. This could inform future design of more targeted therapies against aberrant sialic acid expression. The most critical future research, however, is the need to investigate the synergistic benefits of the vaccine when combined with other immunotherapies or targeted treatments. This is of the utmost importance for moving this approach forward in order to generate the meaningful anti-tumor responses needed to provide maximal benefit to patients, as the influenza vaccine alone was insufficient to eradicate breast cancer in our investigations.

In conclusion, our research demonstrates a promising strategy for future research in breast cancer treatment by leveraging the intratumoral influenza vaccine’s therapeutic activity. We have outlined vital factors for the proper application of this approach and identified key foundational mechanisms of action at play. This approach could provide meaningful benefits in the fight against cancer through further research into combination strategies and patient selection.

## 4. Materials and Methods

### 4.1. Cell Lines and Reagents

Breast cancer cell lines 4T1 and MDA-MB-231 were procured from the American Type Culture Collection (ATCC, Manassas, VA, USA). Cells were thawed, centrifuged, and expanded in recommended culture media as directed by the manufacturer, then cryopreserved for later expansion and further use. Cells were then cultured in RPMI 1640 medium with L-glutamine and supplemented with 10% fetal bovine serum (FBS) at 37 °C in a humidified atmosphere containing 5% CO_2_. The influenza vaccine (FluVax) used for the study was the 2020/2021 season FLUARIX, manufactured by GlaxoSmithKline Biologicals (Dresden, Germany) and stored at 4 °C until use.

### 4.2. Animal Models

Female BALB/c mice, aged 6–8 weeks, were purchased from the Jackson Laboratory. Upon arrival, mice were acclimatized for at least one week under specific pathogen-free conditions with a 12 h light/dark cycle. They had ad libitum access to sterilized food and water. All animal experiments were conducted in accordance with procedures approved by the Institutional Animal Care and Use Committee (IACUC) at Rush University Medical Center.

### 4.3. Tumor Challenge and Vaccine Treatment

For tumor induction, BALB/c mice were anesthetized with isoflurane and administered 200,000 to 500,000 4T1 triple-negative breast cancer cells (ATCC) in the mammary fat pad (MFP) in 100 µL of sterile PBS. Tumor dimensions were measured at least twice weekly using calipers, and tumor volume was calculated using the following formula: volume = (4/3) × 3.1415 × (L/2) × (W/2)^2^. Upon tumors reaching a palpable size of 10–25 mm^2^, mice were separated into treatment groups. Intratumoral injections of 50 µL of each treatment were administered per mouse at each specified time point. Control groups received equivalent volumes of PBS. IL-4 blockade was performed by formulating 300 μg of anti-mouse IL-4 (BioXCell, 11B11) in 500 µL PBS and administering 50 µL of the mix intratumorally per mouse. On days of co-administration with FluVax, this was formulated into one 100 µL injection with FluVax to prevent the spillage that can occur with multiple sequential injections. The influenza vaccine (FLUARIX) was heat-inactivated (Hi-FluVax) by incubating it at 90 °C for 30 min on an IncuBlock Plus heat block (Denville Scientific, Metuchen, NJ, USA) and storing it at 4 °C. Synthetic N-Acetylneuraminic acid (Sigma Aldrich, St. Louis, MO, USA) was mixed directly with influenza vaccines at 4 mg/mL and provided in 50 µL doses as described. Mice were weighed using a digital scale calibrated before each weighing session at a standardized time in the afternoon.

### 4.4. RNA Expression Analysis with NanoString Exhaustion Panel

Total RNA was extracted from tumor tissues using the RNeasy Mini Kit. RNA integrity was assessed using Bioanalyzer RNA kits. The NanoString nCounter Analysis System was employed to analyze gene expression using the Murine NanoString Exhaustion Panel (Seattle, WA, USA). Samples were prepared as per the manufacturer’s instructions, and data acquisition was performed on the nCounter Digital Analyzer. Data normalization was performed using built-in positive and negative controls, and the expression levels of target genes were analyzed relative to housekeeping genes. Additionally, cell type scores were determined and depicted in a heatmap based on the NanoString Cell Type Profiling Module.

### 4.5. Hemagglutination Assay

The hemagglutination assay was performed to assess the binding ability of influenza hemagglutinin to red blood cells (RBCs). Serial dilutions of the influenza vaccine or modified vaccine were prepared in PBS in a U-bottom 96-well plate. Twenty microliters of a 2.5% suspension of RBCs were added to each well. The plate was gently mixed by tapping and incubated at room temperature for one hour in a dark environment. Hemagglutination was visually assessed and photographed.

### 4.6. Cytokine Profiling

Tumor tissues were harvested at the study endpoint, weighed, and homogenized using a gentleMACS Octo homogenizer (Miltenyi Biotec, Bergisch Gladbach, Germany) dissociated in HBSS. The homogenates were centrifuged, and the supernatants were subjected to cytokine profiling using a Custom 20-Plex ProcartaPlex Panel (ThermoFisher Waltham, MA, USA) per manufacturer instructions. The cytokine concentrations were calculated using a 5-parametric curve fit using xPONENT, version 4.03 (Luminex, Austin, TX, USA) in a blinded fashion with measurements performed with the FlexMAP 3D system (Luminex).

### 4.7. Sialic Acid–Hemagglutinin Binding Assay by Flow Cytometry

For the assessment of tumoral sialic acid expression and hemagglutinin’s binding ability to tumor sialic acid residues, a flow cytometry-based binding assay was utilized. Tumor cell sialic acid was blocked using the sialyltransferase inhibitor P-3Fax-Neu5Ac cultured for 48–72 h in 128 µmol/L. Tumor cells were washed and incubated for 45 min at 4 °C with biotinylated MALII (2 mg/mL), SNA (1 mg/mL), or his-tag hemagglutinin (100 ug/mL in an equal distribution of A/Guangdong-Maonan/SWL1536/2019 (H1N1) CNIC-1909, A/Hong Kong/2671/2019 (H3N2), B/Washington/02/2019 (B-Victoria lineage), and B/Phuket/3073/2013 (B-Yamagata lineage). Next, cells were washed and incubated for 40 min at 4 °C with either streptavidin–PE5.5cy or His Tag Antibody AF700 (AD1.1). Cells were subsequently rewashed twice and resuspended in PBS for flow cytometry analysis. Cells were gated FSC/SSC before analysis of fluorophore. Flow cytometry was conducted using a BD LSR2 cytometer. Graphing of flow cytometric data was performed using Floreada.io.

### 4.8. Statistical Analysis

One-way ANOVA with Tukey correction was used to determine statistical significance for data comparisons of multiple groups at a single time point. Two-way ANOVA or mixed-effects model Tukey correction was used to determine statistical significance for data comparisons with multiple time points and groups. The Mantel–Cox log-rank test was performed to determine statistical significance for comparing survival curves. Statistical significance is denoted as ns, not significant, * *p* < 0.05, ** *p* < 0.01, *** *p* < 0.001, and **** *p* < 0.0001. All statistical analyses, including graphical representations, were performed using GraphPad Prism version 9.

## Figures and Tables

**Figure 1 ijms-25-00225-f001:**
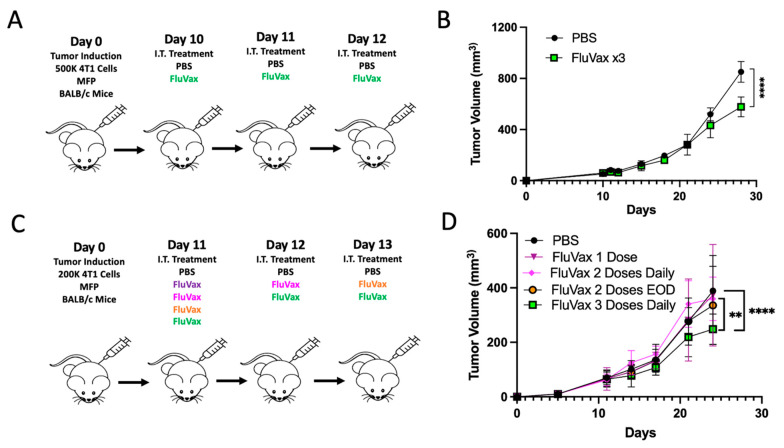
Intratumoral Administration of Flu Vaccine and Its Effect on 4T1 Breast Tumor Growth. (**A**) Experimental design, illustrating the treatment regimen for murine models with 4T1 breast tumors; *n* = 10 mice/group. (**B**) Growth curves for 4T1 breast tumors post-treatment with PBS and three consecutive daily doses of FluVax. (**C**) Modified experimental treatment strategy for examining reduced treatment. (**D**) Comparative analysis of tumor volumes across varying dosages of FluVax treatments versus PBS control. ** *p* < 0.01, **** *p* < 0.0001 (two-way ANOVA with Tukey correction). Error bars represent mean ± SEM. i.t., intratumoral; EOD, every other day; MFP, mammary fat pad injection.

**Figure 2 ijms-25-00225-f002:**
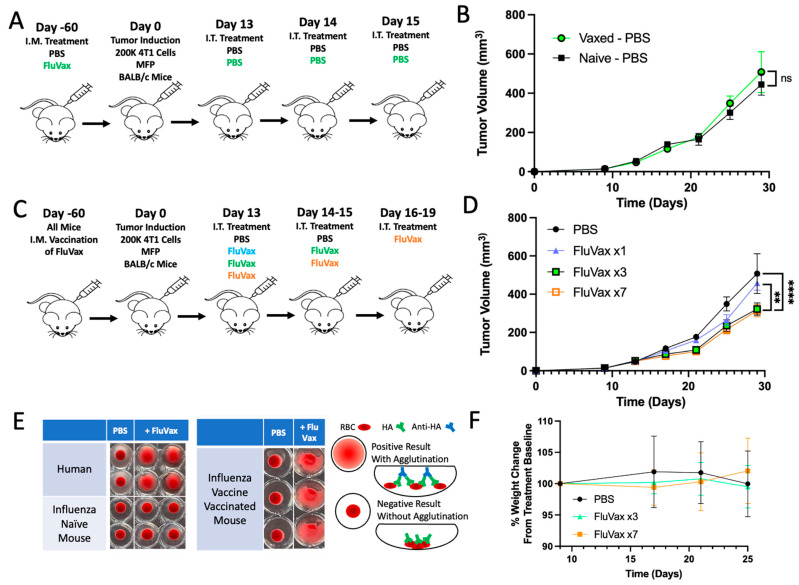
Influence of Prior Vaccination on Tumor Growth and Efficacy of Repeat Intratumoral Influenza Vaccine Administrations. (**A**) Experimental design: Mice underwent prior vaccination or PBS control 60 days pre-4T1 tumor induction; *n* = 10 mice/group. (**B**) Tumor growth trajectories of the experiment are as described in (**A**). No significant (ns) difference was observed between the Flu Vaccinated + PBS IT and Vaccine Naive + PBS IT groups. (**C**) Experimental design: post-prior vaccination, mice received varied intratumoral influenza vaccine doses. *n* = 10 mice/group. (**D**) Tumor growth analysis from the procedure detailed in C. Notable reduction in tumor volume observed in the Vaxed-FluVax ×3 and FluVax ×7 regimens, with both showing a marked decline compared with PBS (**** *p* < 0.01). (**E**) Hemagglutination assay to determine seroconversion in pre-vaccinated mice. Hemagglutination patterns for adult human, young influenza-naive mice, and aged vaccinated mouse samples. Positive agglutination in the vaccinated mouse indicates successful vaccination. (**F**) Percentage weight change from baseline over time in mice following intratumoral administration of PBS, FluVax ×3, and FluVax ×7, examining the systemic effects of repeat influenza vaccine dosing on body weight. ** *p* < 0.01, **** *p* < 0.0001; “ns” symbolizes non-significance (two-way ANOVA with Tukey correction). Error bars: mean ± SEM. i.t., intratumoral; MFP, mammary fat pad injection.

**Figure 3 ijms-25-00225-f003:**
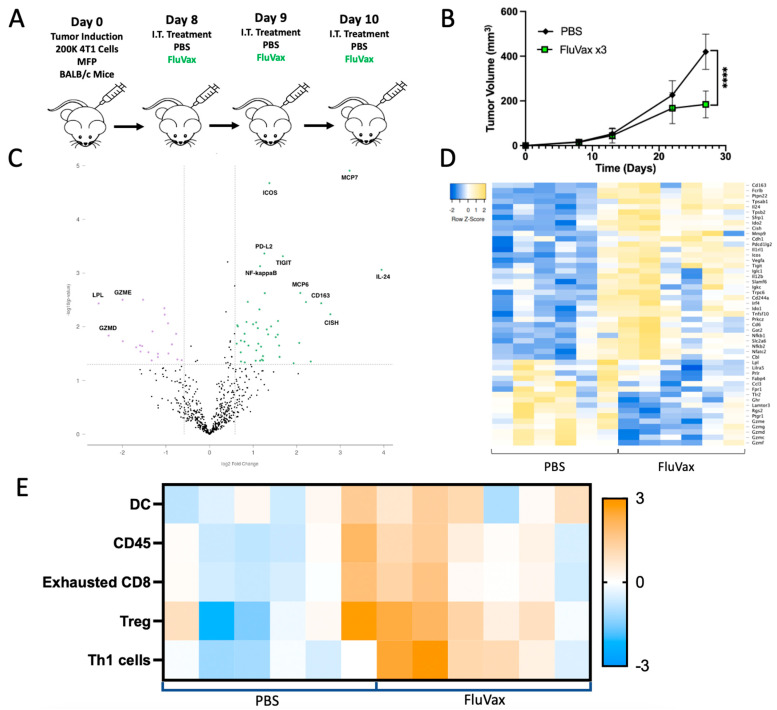
Intratumoral Influenza Vaccine Administration Modulates the 4T1 Tumor Microenvironment. (**A**) An experimental design depicting the treatment timeline with tumor tissue harvest on day 27; *n* = 10 mice/group. (**B**) Tumor growth kinetics of the experiment described in (**A**). (**C**) Volcano plot representing differentially expressed genes from the NanoString exhaustion panel, with genes of notable significance labeled. The x-axis denotes the log2 fold change, whereas the *y*-axis represents the −log10 (*p*-value). Compared to controls, upregulated (green dots) and downregulated (purple dots) genes in the FluVax-treated group are visualized as dots on the right and left of the plot, respectively. (**D**) Heatmap showcasing z-score normalized expression levels of select genes from the NanoString nCounter Murine Exhaustion Panel. Columns represent individual samples from either PBS-treated (left) or FluVax-treated (right) mice. Shades of blue indicate reduced expression, while shades of yellow signify enhanced expression relative to the mean. (**E**) Two-color heatmap of cell type profiling from the NanoString Murine Exhaustion Panel and Cell Type Profiling Module. Shades of blue indicate reduced expression, while shades of orange signify enhanced expression relative to the mean. **** *p* < 0.001 (two-way ANOVA with Tukey correction). Error bars: mean ± SEM. i.t., intratumoral; MFP, mammary fat pad injection.

**Figure 4 ijms-25-00225-f004:**
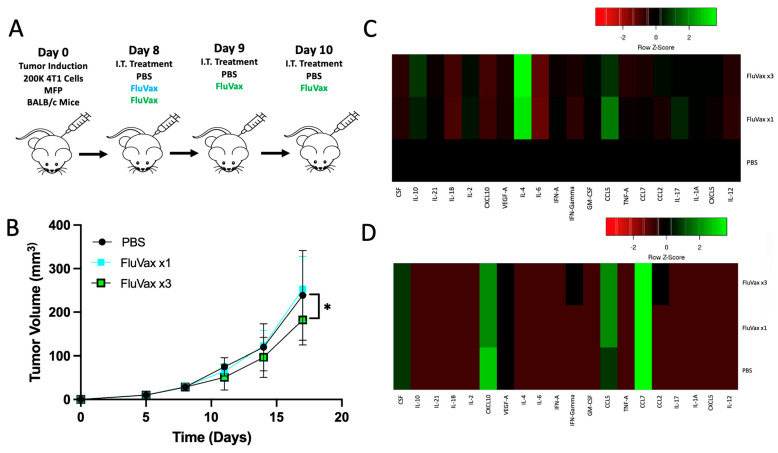
Evaluation of FluVax Treatment and Its Modulation of Tumor Microenvironment Cytokines. (**A**) Schematic representation of the experimental timeline. (**B**) Analysis of tumor volume until tissue harvest; *n* = 10 mice/group. (**C**,**D**) Luminex multiplex heatmap panels displayed as fold change (**C**) and total quantification (**D**) of tumor samples from B. (**E**) In-depth quantification of intratumoral cytokine concentrations, spotlighting IL-1b, IL-6, and IL-4. (**F**) Treatment design, illustrating the sequence of FluVax and anti-IL-4 antibody interventions in 4T1 tumor-bearing mice; *n* = 10 mice/group. (**G**) Tumor volume growth curve, examining effects of IL-4 blocking in combination with FluVax. (**H**) Kaplan–Meier survival curves, portraying impact on survival with FluVax ×3 treatment with and without anti-IL-4 interventions. * *p* < 0.05, ** *p* < 0.01, *** *p* < 0.001, **** *p* < 0.0001; “ns” symbolizes non-significance (two-way ANOVA with Tukey correction (B & G), Mantel–Cox log-rank test (C), one-way ANOVA with Tukey correction (E)). Error bars: mean ± SEM. i.t., intratumoral; MFP, mammary fat pad injection.

**Figure 5 ijms-25-00225-f005:**
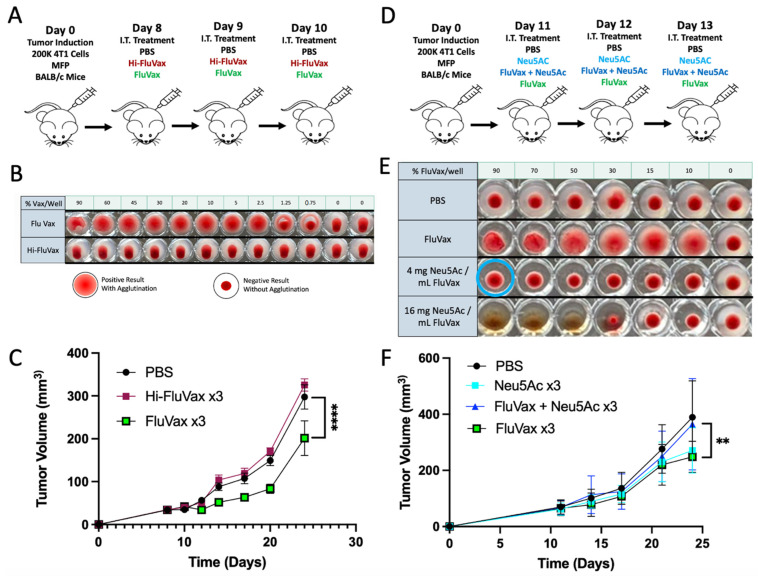
Impact of Modified Flu Vaccine Binding Ability on Anti-Tumor Responses. (**A**) Treatment protocol detailing intratumoral injections of PBS, Hi-FluVax, and FluVax over consecutive days; *n* = 10 mice/group. (**B**) Hemagglutination assay visualizing the binding capacity of both FluVax and Hi-FluVax across varying concentrations, highlighting the decline in binding ability with increased FluVax dilution. Effective binding is marked by positive agglutination. (**C**) Tumor volume growth curve, differentiating the therapeutic effects of triple doses of FluVax against Hi-FluVax. (**D**) Experimental design and treatment regimen for the subsequent group of mice using synthetic Neu5AC in a mixture with FluVax to block sialic acid binding; *n* = 10 mice/group. (**E**) Hemagglutination assay outcomes under various conditions: control (PBS), standard FluVax, and combinations of FluVax with two distinct concentrations of N-acetylneuraminic acid (Neu5Ac)—a competitive binding agent of the hemagglutinin binding site. The blue circle indicates optimal titration at full vaccine concentration used in vivo (**F**). (**F**) Growth dynamics of tumors demonstrating reduced efficacy with combination treatment. ** *p* < 0.01, **** *p* < 0.0001 (two-way ANOVA with Tukey correction C and F). Error bars: mean ± SEM. i.t., intratumoral; MFP, mammary fat pad injection.

**Figure 6 ijms-25-00225-f006:**
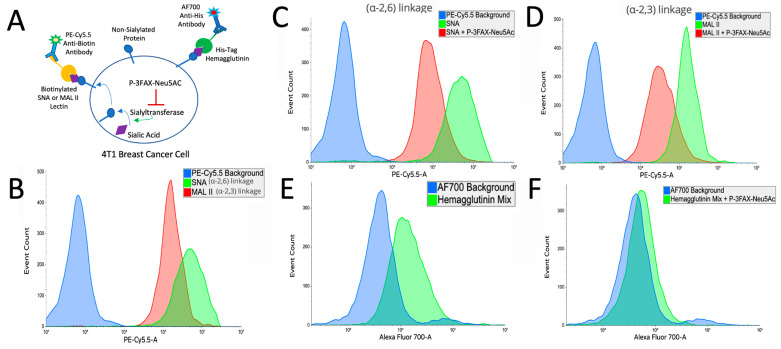
Sialic Acid Expression Modulates Hemagglutinin Binding in 4T1 Tumor Model. (**A**) Diagram illustrating the treatment of 4T1 breast cancer cells with the compound P-3FAX-Neu5Ac used to downregulate the sialic acid expression of these cells. Biotinylated SNA or MAL II lectins were used to detect specific sialic acid linkages, while AF700 anti-His antibody was employed to identify His-Tag Hemagglutinin binding. (**B**) Comparative analysis of sialic acid expression of two distinct sialic acid linkages α2,6 and α2,3, on 4T1 cells. The binding affinities were assessed using biotinylated lectins SNA for the α2,6 linkage and MAL II for the α2,3 linkage. The PECy5.5 background indicates the background fluorescence intensity without specific lectin binding. (**C**,**D**) The examination of sialic downregulation by P-3FAX-Neu5Ac on sialic acid linkages α2,6 (**C**) and α-2,3 (**D**). (**E**,**F**) The binding efficacy of His-tagged hemagglutinin to sialic acid-expressing 4T1 cells is dependent on sialic acid expression. The binding affinities were assessed using His-tagged hemagglutinin. The AF700 background indicates the background fluorescence intensity without hemagglutinin binding. The loss of hemagglutinin binding on cells with downregulated sialic acid expression due to treatment with P-3FAX-Neu5Ac can be seen by the difference in binding with (**E**) P-3FAX-Neu5Ac treatment and (**F**) without.

**Figure 7 ijms-25-00225-f007:**
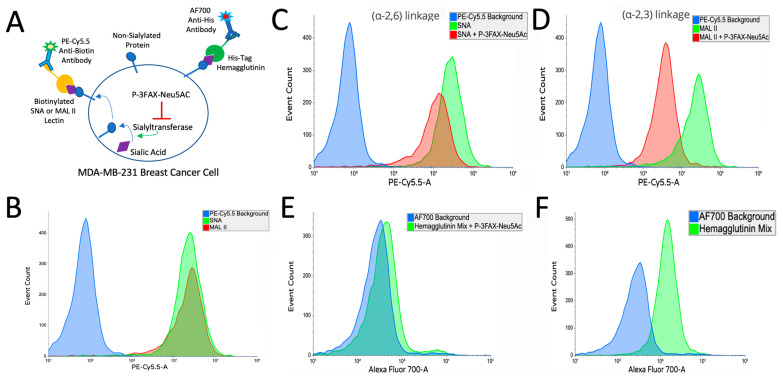
Sialic Acid Expression Modulates Hemagglutinin Binding in Human MDA-MB-231 Breast Cancer Model. (**A**) Diagram illustrating the treatment of MDA-MB-231 breast cancer cells with the compound P-3FAX-Neu5Ac used to downregulate the sialic acid expression of these cells. Biotinylated SNA or MAL II lectins were used to detect specific sialic acid linkages, while AF700 anti-His antibody was employed to identify His-Tag Hemagglutinin binding. (**B**) Comparative analysis of sialic acid expression of two distinct sialic acid linkages, α2,6 and α2,3, on 4T1 cells. The binding affinities were assessed using biotinylated lectins SNA for the α2,6 linkage and MAL II for the α2,3 linkage. The PECy5.5 background indicates the background fluorescence intensity without specific lectin binding. (**C**,**D**) The examination of sialic downregulation by P-3FAX-Neu5Ac on sialic acid linkages α2,6 (**C**) and α-2,3 (**D**). (**E**,**F**) The binding efficacy of His-tagged hemagglutinin to sialic acid-expressing MDA-MB-231 cells is dependent on sialic acid expression. The binding affinities were assessed using his-tagged hemagglutinin. The AF700 background indicates the background fluorescence intensity without hemagglutinin binding. The loss of hemagglutinin binding on cells with downregulated sialic acid expression due to treatment with P-3FAX-Neu5Ac can be seen by the difference in binding with (**E**) P-3FAX-Neu5Ac treatment and (**F**) without.

## Data Availability

Data available in the publicly accessible repository GEO: with the accession number: GSE251825.

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
