# Peer review of "Intratumoral Influenza Vaccine Administration Attenuates Breast Cancer Growth and Restructures the Tumor Microenvironment through Sialic Acid Binding of Vaccine Hemagglutinin"

_ijms, 2023, doi:10.3390/ijms25010225_

Round 1

Reviewer 1 Report

Comments and Suggestions for Authors

Daniels et al. set out to test the anti-tumour effect of intratumoral influenza vaccine administration on the 4T1 breast cancer model and to understand the mechanisms behind this. Whilst the paper is well-written I have several concerns regarding the authors conclusions from the limited data provided.

The effect of intratumoral administration of the Flu vaccine in the 4T1 breast tumour growth model was not particularly impressive. Whilst it did attenuate tumour growth no cures of the mice were seen which are often seen with the use of e.g. oncolytic viruses that also have a powerful effect on changing the immune landscape within tumours.

The authors state that intratumoural  influenza vaccine administration reshapes the 4T1 tumour microenvironment. This is based on the RNA expression evaluated using the murine nCounter Immune exhaustion panel. The only data the authors show from this panel is a volcano plot of differentially expressed genes, the results of which they use to infer T cell influx. Why did the authors not carry out immune cell deconvolution from the immune exhaustion panel data and show the immune cell profiles in their PBS and FluVax cohorts to back up this statement? Furthermore, was there any tissue from the tumours available for IHC analysis of immune infiltration? From the heatmap of normalized expression levels of select genes it would appear that granzyme genes are downregulated in the FluVax group compared to the PBS cohort suggesting that you are not getting a cytotoxic T cell response.

Upon looking at the heatmap of normalized expression levels of select genes, there is also a noticeable upregulation of immunosuppressive genes such as IDO1 and 2, VEGFA, and immune checkpoint ligands PD-L2 and TIGIT along with CD163 indicative of a more M2 (immunosuppressive)-like macrophage population. This profile would suggest that intratumoural  influenza vaccine administration is not inducing an anti-tumour immune response which may also explain the limited treatment efficacy.

The authors then go on to propose that as IL-4 was the only differentially expressed cytokine between single and repeat administrations of influenza vaccine, it must be mediating an anti-tumour response. However, considerable clinical evidence suggests that IL4 can act as a tumor-promoting molecule, as it is found at high levels in multiples types of human primary and metastatic cancers. IL4 can also suppress tumor immunity in several ways e.g. downregulating the development of Th1 immunity and acting on CD8 T cells to render them noncytotoxic which appears to fit with the current study’s results of down-regulated granzyme expression in the FluVax group.

Whilst the authors did attempt to show that blocking IL-4 resulted in the anti-tumour benefit of their influenza vaccine being eliminated, this was only shown at one time point in their tumour growth curve and only resulted in a marginal effect difference in survival.

The authors then go on to propose that the anti-tumor effects of IL-4 are in part mediated through the upregulation of IL-24. In the discussion the authors state that the unique characteristics of IL-24 are to induce apoptosis,  inhibit angiogenesis and increase T cell fitness. Was there any evidence of increased apoptosis in the FluVax cohort compared to PBS from the murine nCounter Immune exhaustion panel data? It would have been informative to look within the tumour tissues for evidence of tumour cell death or changes in angiogenesis.

Due to the lack of data provided (above comments) to back up the proposed mechanisms of the tumour growth attenuation seen in response to FluVax I do not think the authors can say that they have identified key foundational mechanisms of action at play. Also I do not think the data backs up the authors statement in their abstract that the vaccine facilitated crucial shifts to an anti-cancer tumour immune signature.

Reviewer 2 Report

Comments and Suggestions for Authors

The work by Daniels and colleagues provides an interesting and convincing presentation on the antitumor efficacy of anti influenza vaccine against breast tumor, in the triple negative 4T1 mouse model.

Tumor growth data are well presented and convincing, and the molecular analysis leading to IL4 and IL24 pathway is sound.

Nevertheless, since the authors claim a reshaping of tumor microenvironment and recall IL4-mediated immune response as a mechanisms behind vaccine efficacy, the reviewer and possible readers may wonder why no data on antitumor immune response against 4T1 tumors was reported. Are there any information about tumor mass infiltration by immune cells or the induction of an antitumor immune response, beside cytokine modulation?

Please, include such information on the paper, if available.

If this is not the case, the discussion section should be reconsidered and a more speculative attitude should be used, when it comes to immune-related mechanisms of action.

Round 2

Reviewer 1 Report

Comments and Suggestions for Authors

I am satisfied with all the efforts made to address my comments.

Author Response

See PDF attached
